# Thromboembolic Events in Users of Warfarin Treated with Different Skeletal Muscle Relaxants

**DOI:** 10.3390/medicina58091171

**Published:** 2022-08-29

**Authors:** Charles E. Leonard, Colleen M. Brensinger, Warren B. Bilker, Samantha E. Soprano, Neil Dhopeshwarkar, Todd E. H. Hecht, Scott E. Kasner, Edith A. Nutescu, Anne Holbrook, Matthew Carr, Darren M. Ashcroft, Cheng Chen, Sean Hennessy

**Affiliations:** 1Center for Real-World Effectiveness and Safety of Therapeutics, Perelman School of Medicine, University of Pennsylvania, Philadelphia, PA 19104, USA; 2Department of Biostatistics, Epidemiology, and Informatics, Perelman School of Medicine, University of Pennsylvania, Philadelphia, PA 19104, USA; 3Neuropsychiatry Section, Department of Psychiatry, Perelman School of Medicine, University of Pennsylvania, Philadelphia, PA 19104, USA; 4Division of General Internal Medicine, Department of Medicine, Perelman School of Medicine, University of Pennsylvania, Philadelphia, PA 19104, USA; 5Division of Vascular Neurology, Department of Neurology, Perelman School of Medicine, University of Pennsylvania, Philadelphia, PA 19104, USA; 6Department of Pharmacy Practice and Center for Pharmacoepidemiology and Pharmacoeconomic Research, College of Pharmacy, University of Illinois at Chicago, Chicago, IL 60612, USA; 7Division of Clinical Pharmacology and Toxicology, Department of Medicine, Faculty of Health Sciences, McMaster University, Hamilton, ON L8S 4L8, Canada; 8Center for Pharmacoepidemiology and Drug Safety, Division of Pharmacy & Optometry, School of Health Sciences, University of Manchester, Manchester M13, UK; 9Department of Systems Pharmacology and Translational Therapeutics, Perelman School of Medicine, University of Pennsylvania, Philadelphia, PA 19104, USA

**Keywords:** central muscle relaxants, drug interactions, Medicaid, pharmacoepidemiology, thromboembolism, warfarin

## Abstract

*Background and Objectives*: Warfarin and a skeletal muscle relaxant are co-treatments in nearly a quarter-million annual United States (US) office visits. Despite international calls to minimize patient harm arising from anticoagulant drug interactions, scant data exist on clinical outcomes in real-world populations. We examined effects of concomitant use of warfarin and individual muscle relaxants on rates of hospitalization for thromboembolism among economically disadvantaged persons. *Materials and Methods*: Using 1999–2012 administrative data of four US state Medicaid programs, we conducted 16 retrospective self-controlled case series studies: half included concomitant users of warfarin + one of eight muscle relaxants; half included concomitant users of an inhaled corticosteroid (ICS) + one of eight muscle relaxants. The ICS analyses served as negative control comparisons. In each study, we calculated incidence rate ratios (IRRs) comparing thromboembolism rates in the co-exposed versus warfarin/ICS-only exposed person-time, adjusting for time-varying confounders. *Results*: Among ~70 million persons, we identified 8693 warfarin-treated subjects who concomitantly used a muscle relaxant, were hospitalized for thromboembolism, and met all other inclusion criteria. Time-varying confounder-adjusted IRRs ranged from 0.31 (95% confidence interval: 0.13–0.77) for metaxalone to 3.44 (95% confidence interval: 1.53–7.78) for tizanidine. The tizanidine finding was robust after quantitatively adjusting for negative control ICS findings, and in numerous prespecified secondary analyses. *Conclusions*: We identified a potential >3-fold increase in the rate of hospitalized thromboembolism in concomitant users of warfarin + tizanidine vs. warfarin alone. Alternative explanations for this finding include confounding by indication, a native effect of tizanidine, or chance.

## 1. Introduction

Minimizing patient harm associated with anticoagulants and their drug interactions is an international patient safety goal. To address knowledge gaps in anticoagulant safety, the United States (US) Department of Health and Human Services issued a call to generate real-world evidence on anticoagulant drug interactions [1]. Yet, the evidence base underlying many anticoagulant interactions is limited [2,3]. Most evidence arises from case reports and pharmacokinetic studies. Of the few population-based interaction studies of a clinical endpoint, most have investigated bleeding from over-anticoagulation. However, real-world evidence on anticoagulant interactions and thrombotic consequences of under-anticoagulation is limited.

Recent data suggest that warfarin and a skeletal muscle relaxant are co-treatments in 240,000 annual US office visits [4]. The scale of co-treatment is not surprising since warfarin use remains common and muscle relaxant use is increasing [5] as providers seek alternatives to opioids. The potential for drug interactions between warfarin and muscle relaxants has received little attention because most muscle relaxants are neither metabolized by nor inhibit cytochrome P450 (CYP) 2C9 [6], the isozyme primarily responsible for warfarin’s hepatic metabolism; further, it has been 50 years since pharmacokinetic studies were conducted in concomitant users [7,8]. Yet, recent hypothesis-free screening for anticoagulant interactions generated potential signals of decreased international normalized ratios (INRs) among concomitant users of warfarin and some muscle relaxants [9]. In response, we conducted a series of hypothesis-testing pharmacoepidemiologic studies to generate real-world evidence on these drug interactions. We specifically examined the effects of concomitant use of warfarin and individual muscle relaxants on rates of thromboembolism, i.e., venous thromboembolism and ischemic stroke—consequences of under-anticoagulation—among economically disadvantaged persons, a population especially vulnerable to adverse drug events [1].

## 2. Materials and Methods

We conducted 16 retrospective self-controlled case series (SCCS) studies: half included concomitant users of warfarin + one of eight muscle relaxants; half included concomitant users of an inhaled corticosteroid (ICS) + one of eight muscle relaxants. The ICS analyses served as negative control comparisons [10]. We defined exposure by the presence/absence of muscle relaxant therapy, based on prescription dispensing dates and days’ supplied, on each eligible observation day. We defined the outcome as hospitalization for thromboembolism (i.e., venous thromboembolism, ischemic stroke). We adjusted for numerous time-varying covariates, assessed on each observation day. Time-invariant factors are inherently accounted for by the self-controlled nature of the SCCS design. This substantial benefit is accompanied by reliance on the following assumptions: outcomes are independent or rare; outcomes do not appreciably affect observation time or subsequent exposure; and exposures do not affect outcome ascertainment. We conducted analyses within 1999–2012 Medicaid data from California, Florida, New York, and Pennsylvania, linked to Medicare data for dual-eligibles and the Social Security Administration Death Master File (Appendix A), which does not include laboratory results such as the international normalized ratio (INR). We calculated incidence rate ratios (IRRs), in which thromboembolism rates in co-exposed and warfarin-only exposed persons were in the numerator and denominator, respectively. We conducted analyses using SAS version 9.4 (SAS Institute Inc.: Cary, NC, USA). The University of Pennsylvania’s institutional review board approved this research via expedited procedure set forth in 45 CFR 46.110. See Appendix A for further detail on the study design.

## 3. Results

Among ~70 million Medicaid beneficiaries in states contributing data, we identified 8693 warfarin-treated subjects who concomitantly used a muscle relaxant, experienced ≥1 thromboembolism event, and met all other inclusion criteria. Subjects were predominantly female (67.4%), white (45.6%), with a median age of 67.4 years, and contributed 1,005,246 observation days. The mean per-subject observation period was 116 days. Table 1 further describes subjects. We did not examine chlorzoxazone or orphenadrine, as these samples contained <10 persons. 

Confounder-adjusted IRRs for thromboembolism ranged from 0.31 (95% confidence interval 0.13–0.77) for *warfarin + metaxalone* to 3.44 (1.53–7.78) for *warfarin + tizanidine*. Negative control findings ranged from 0.39 (0.10–1.49) for *ICS + tizanidine* to 1.99 (1.04–3.83) for *ICS + carisoprodol*. See Appendix A and Figure 1. Findings from secondary analyses (Appendix A) were consistent with primary analyses.

## 4. Discussion

Warfarin users are commonly co-treated with a muscle relaxant [4], and tizanidine prescribing is specifically on the rise [11]. Further, a previous hypothesis-free screening study suggested that *warfarin + tizanidine* use may result in a modest, delayed INR reduction (−0.4 units during the third and fourth months of concomitant use) [9]. We therefore used population-based data to examine the association between concomitant use of warfarin with different muscle relaxants and thromboembolism—principally finding a >3-fold increase in the rate of hospitalization for thromboembolism among concomitant users of *warfarin + tizanidine* vs. warfarin alone. This finding, if causal, may represent a clinically relevant drug interaction, yet could also be explained by: confounding by indication for tizanidine; a native pharmacodynamic effect of tizanidine; the failure of an assumption underlying the statistical (i.e., SCCS) model; and/or chance. We subsequently discuss the possibilities of causation, systematic error, and chance. 

A plausible pharmacokinetic mechanism would help support a case for causality. While *R*-warfarin and tizanidine are CYP1A2 substrates, we are unaware of evidence that tizanidine induces CYP1A2, which would enhance *R*-warfarin inactivation. We therefore must consider potential non-causal explanations, such as confounding. As an example, initiation of tizanidine to treat spasticity may portend a multiple sclerosis (MS) flare, and MS is associated with thromboembolism [12,13]. However, this explanation seems unlikely since our finding for baclofen, also used in MS, was null (IRR: 0.81, 0.53–1.25). That said, tizanidine may be used in settings of more severe MS-associated spasticity [14], and immobility accompanying more severe MS may portend a high thromboembolism risk period [15]. Another example of confounding would be the common co-occurrence of muscle relaxant use and smoking [16]. Smoking, poorly measured in our data, may induce CYP1A2 resulting in enhanced *R-*warfarin inactivation. This explanation is unlikely since: within-subject smoking status is unlikely to change over the relatively short observation time; and our finding for cyclobenzaprine, also a CYP1A2 substrate, was null (IRR: 0.95, 0.69–1.32). Another potential non-causal explanation is a native pharmacodynamic effect of tizanidine. This is unlikely since: tizanidine’s alpha-2 agonist effects do not affect stroke [17]; and our negative control analysis of *ICS* + *tizanidine* was inconsistent with a large positive association (upper 95% confidence limit: 1.49). It is additionally important to note that prior screening findings [9] identified: similar INR reductions for tizanidine and for carisoprodol, yet our thromboembolism finding for carisoprodol was consistent with the null; and an INR increase for metaxalone, consistent with our protective thromboembolism finding. Further, prior screening [9] did not produce a thromboembolism signal for tizanidine. 

We must also consider the potential for systematic error introduced by violations of assumptions underlying the SCCS method. *First*, recurrent outcomes must be independent. This assumption may be violated if different within-patient factors cause a subsequent vs. initial thromboembolic event. Yet, our tizanidine finding held in a secondary analysis limited to persons with one thromboembolic event (IRR: 3.07, 1.24–7.60). *Second*, outcomes must not affect observation time or subsequent exposure. The former may be violated given thromboembolism-related mortality; yet, our tizanidine finding held in a secondary analysis limited to persons surviving during observation (IRR: 3.42, 1.51–7.74). The latter may be violated since muscle relaxants may be initiated after a stroke to treat spasticity; yet, this would explain a protective (not elevated) finding, as such stroke events would occur during warfarin-only exposed observation days and thereby artificially inflate the IRR’s denominator. *Third*, exposures must not affect outcome ascertainment. We cannot think of a mechanism by which tizanidine could influence the diagnosis of or billing for thromboembolism. An additional limitation includes our lack of access to INR results. Finally, we cannot rule out the possibility of chance findings.

## 5. Conclusions

Using an epidemiologic design that inherently controls for measured and unmeasured static confounders, adjusting for critical time-varying confounders, and using a negative control, we identified a potential >3-fold increase in the rate of thromboembolism in concomitant users of *warfarin + tizanidine* vs. warfarin alone. While this finding may represent a clinically relevant drug interaction, alternative explanations include confounding, bias, and chance. Future work using a different data source (especially one with access to INR values) and/or alternative study design should seek to replicate this finding.

## Figures and Tables

**Figure 1 medicina-58-01171-f001:**
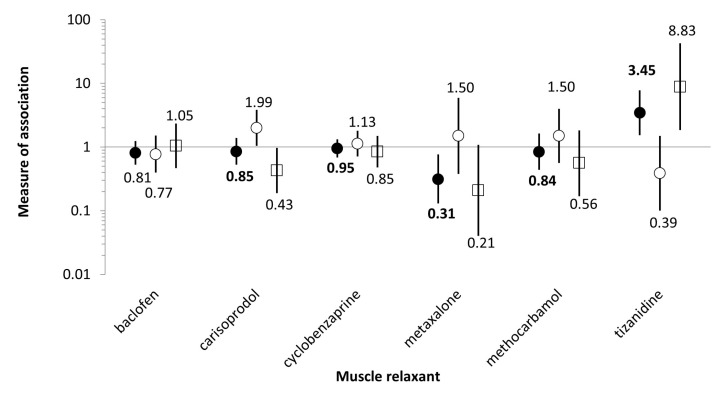
Confounder-adjusted incidence rate ratios (IRRs, circles) and ratios of adjusted incidence rate ratios (rIRRs, squares) with 95% confidence intervals for thromboembolism, for primary analyses, by muscle relaxant precipitant drug. *Legend*: Black circles: confounder-adjusted incidence rate ratios (IRRs) for warfarin (bolded since effect estimates of primary interest); White circles: confounder adjusted IRRs for inhaled corticosteroids (negative control); White squares: ratios of IRRs, i.e., ([adjusted incidence rate ratios for use of warfarin + muscle relaxant vs. warfarin alone]/[adjusted incidence rate ratios for use of inhaled corticosteroid + muscle relaxant vs. inhaled corticosteroid alone]).

**Table 1 medicina-58-01171-t001:** Characteristics of persons under study who, by nature of the self-controlled study design, experienced at least one thromboembolic outcome during treatment with the object drug.

		Object Drug	
		Warfarin	Inhaled Corticosteroid (Negative Control)
** *Persons, person-days, and outcome occurrence* **			
Persons, total		8693	4582
Person-days of observation time, median per individual (Q1–Q3)		67.0 (36.0–134.0)	53.0 (36.0–109.0)
Person-days of observation time, total		1,005,246	521,722
Exposed to a skeletal muscle relaxant		42,572 (4.2%)	38,351 (7.4%)
Unexposed to a skeletal muscle relaxant		962,674 (95.8%)	483,371 (92.6%)
Thromboembolism outcomes during observation time		9396	4662
Exposed to a skeletal muscle relaxant		474 (5.0%)	318 (6.8%)
Unexposed to a skeletal muscle relaxant		8922 (95.0%)	4344 (93.2%)
Thromboembolism outcomes during observation time that were venous thromboembolisms, vs. ischemic strokes (% of total thromboembolism outcomes)		6569 (69.9%)	2381 (51.1%)
** *Demographics and other baseline clinical characteristics, at start of observation time* **			
Age, in years, median (Q1–Q3)		67.4 (51.3–78.3)	69.9 (57.4–79.4)
Female		5598 (64.4%)	3052 (66.6%)
Race			
White		3966 (45.6%)	2153 (47.0%)
Black		1877 (21.6%)	915 (20.0%)
Hispanic/Latino		1357 (15.6%)	675 (14.7%)
Other/unknown		1493 (17.2%)	839 (18.3%)
State of residence			
CA		3835 (44.1%)	1885 (41.1%)
FL		1528 (17.6%)	864 (18.9%)
NY		2606 (30.0%)	1369 (29.9%)
PA		724 (8.3%)	464 (10.1%)
Calendar year (see Appendix A)			
1999		287 (3.3%)	43 (0.9%)
2000		512 (5.9%)	79 (1.7%)
2001		564 (6.5%)	127 (2.8%)
2002		602 (6.9%)	187 (4.1%)
2003		625 (7.2%)	251 (5.5%)
2004		531 (6.1%)	260 (5.7%)
2005		613 (7.1%)	295 (6.4%)
2006		728 (8.4%)	406 (8.9%)
2007		663 (7.6%)	355 (7.7%)
2008		653 (7.5%)	423 (9.2%)
2009		799 (9.2%)	524 (11.4%)
2010		774 (8.9%)	547 (11.9%)
2011		683 (7.9%)	599 (13.1%)
2012		659 (7.6%)	486 (10.6%)
Nursing home residence (Yes)		124 (1.4%)	70 (1.5%)
CHA_2_DS_2_-VASc score, median (Q1–Q3)		3.0 (1.0–4.0)	3.0 (1.0–4.0)
** *Exposure to skeletal muscle relaxant, during observation time (day level)* **			
Antispastic Agent	Baclofen	11,794 (1.2%)	10,007 (1.9%)
Antispasmodic Agents	Carisoprodol	11,070 (1.1%)	9611 (1.8%)
	Chlorzoxazone	551 (0.1%)	42 (0.0%)
	Cyclobenzaprine	12,502 (1.2%)	13,382 (2.6%)
	Metaxalone	1739 (0.2%)	1273 (0.2%)
	Methocarbamol	2515 (0.3%)	2102 (0.4%)
	Orphenadrine	176 (0.0%)	239 (0.0%)
Antispastic-Antispasmodic	Tizanidine	2225 (0.2%)	1695 (0.3%)
** *Time-varying covariates, on the current observation day or in the prior 30 days (unless otherwise noted)* **			
*Major non-chronic risk factors for venous thromboembolism*			
Hospital discharge		359,313 (35.7%)	150,271 (28.8%)
Venous thromboembolism in the prior 90 days		327,187 (32.5%)	74,466 (14.3%)
*Major non-chronic risk factor for ischemic stroke*			
Ischemic stroke in the prior 90 days		101,368 (10.1%)	61,226 (11.7%)
*Drug exposures that increase risk of venous thromboembolism and ischemic stroke*			
Oral contraceptive/hormone replacement therapy		15,777 (1.6%)	11,813 (2.3%)
Nonsteroidal anti-inflammatory drug		89,310 (8.9%)	95,203 (18.2%)
Tamoxifen		1475 (0.1%)	522 (0.1%)
Nicotine		5082 (0.5%)	5017 (1.0%)
Recombinant factor VIIa		0 (0.0%)	‡
Cisplatin		1405 (0.1%)	346 (0.1%)
*Drug exposures that increase risk of venous thromboembolism, but not ischemic stroke*			
Testosterone		3661 (0.4%)	3591 (0.7%)
Dexamethasone		9104 (0.9%)	2561 (0.5%)
Methylprednisolone		5900 (0.6%)	8747 (1.7%)
Epoetin alpha/darbepoetin alpha		19,233 (1.9%)	8218 (1.6%)
Filgrastim/sargramostim		4400 (0.4%)	931 (0.2%)
Flutamide		223 (0.0%)	0 (0.0%)
Goserelin		647 (0.1%)	147 (0.0%)
Leuprolide		1153 (0.1%)	407 (0.1%)
Raloxifene		6539 (0.7%)	7082 (1.4%)
Anastrozole		3539 (0.4%)	826 (0.2%)
Megestrol		20,602 (2.0%)	14,633 (2.8%)
Cyclosporine		1279 (0.1%)	360 (0.1%)
Infliximab		385 (0.0%)	316 (0.1%)
Immune globulin		980 (0.1%)	527 (0.1%)
Interferon gamma-1b		24 (0.0%)	0 (0.0%)
Sirolimus/tacrolimus		4173 (0.4%)	1394 (0.3%)
Aldesleukin		52 (0.0%)	0 (0.0%)
Bevacizumab		1159 (0.1%)	342 (0.1%)
Bleomycin		108 (0.0%)	0 (0.0%)
Carboplatin		4359 (0.4%)	1414 (0.3%)
Denileukin		126 (0.0%)	33 (0.0%)
Docetaxel		940 (0.1%)	326 (0.1%)
Estramustine		80 (0.0%)	155 (0.0%)
Fluorouracil		3095 (0.3%)	806 (0.2%)
Imatinib		224 (0.0%)	66 (0.0%)
Irinotecan		1247 (0.1%)	232 (0.0%)
Lenalidomide		699 (0.1%)	66 (0.0%)
Paclitaxel		3611 (0.4%)	1002 (0.2%)
Thalidomide		1970 (0.2%)	0 (0.0%)
Heparin (including low molecular weight heparin)		89,942 (8.9%)	24,034 (4.6%)
Pentosan		60 (0.0%)	64 (0.0%)
Chlorpromazine		2116 (0.2%)	276 (0.1%)
Clozapine		1198 (0.1%)	891 (0.2%)
Olanzapine		20,842 (2.1%)	19,714 (3.8%)
Quetiapine		25,349 (2.5%)	26,212 (5.0%)
Risperidone		20,212 (2.0%)	18,860 (3.6%)
Thioridazine		898 (0.1%)	648 (0.1%)
Celecoxib		26,992 (2.7%)	27,321 (5.2%)
Botulinum toxin		309 (0.0%)	178 (0.0%)
Papaverine		159 (0.0%)	12 (0.0%)
Topiramate		9856 (1.0%)	6773 (1.3%)
*Drug exposures that increase risk of ischemic stroke, but not venous thromboembolism*			
Selective serotonin reuptake inhibitor/serotonin and norepinephrine reuptake inhibitor		161,503 (16.1%)	125,524 (24.1%)
*Disease influencing anticoagulation*			
Acute infection on current day or in prior 14 days		106,105 (10.6%)	67,681 (13.0%)
*Other drug exposures influencing anticoagulation*			
Oral anticoagulant (non-warfarin)		20 (0.0%)	NA
Oral anticoagulant		NA	118,230 (22.7%)
Oral antiplatelet		57,492 (5.7%)	115,997 (22.2%)
Aspirin		50,928 (5.1%)	74,362 (14.3%)
Injectable/subcutaneous anticoagulant		81,923 (8.1%)	20,656 (4.0%)
*Drug exposures related to drug interactions **			
Oral agents that can interact with warfarin **		219,787 (21.9%)	NA
Oral agents that can interact with muscle relaxants **		28,078 (2.8%)	18,172 (3.5%)
CYP2C9 inhibitors †		85,801 (8.5%)	53,407 (10.2%)
CYP2C9 inducers †		24,174 (2.4%)	8229 (1.6%)
CYP1A2 inhibitors †		68,810 (6.8%)	44,959 (8.6%)
CYP1A2 inducers †		12,822 (1.3%)	4905 (0.9%)
CYP2C19 inhibitors †		282,004 (28.1%)	225,690 (43.3%)
CYP2C19 inducers †		8379 (0.8%)	8823 (1.7%)
CYP2D6 inhibitors †		135,687 (13.5%)	97,209 (18.6%)
CYP2E1 inhibitors †		41 (0.0%)	0 (0.0%)
CYP2E1 inducers †		1170 (0.1%)	843 (0.2%)
CYP3A4 inhibitors †		110,361 (11.0%)	71,567 (13.7%)
CYP3A4 inducers †		80,359 (8.0%)	57,269 (11.0%)
CYP2C8 inhibitors †		34,548 (3.4%)	102,645 (19.7%)
CYP2B6 inhibitors †		50,283 (5.0%)	101,917 (19.5%)
CYP2B6 inducers †		61,987 (6.2%)	26,819 (5.1%)
*Other non-chronic factors potentially related to muscle relaxant exposure*			
Diseases of the esophagus (including GERD)		48,237 (4.8%)	33,696 (6.5%)
Disorders of musculoskeletal system and connective tissue		379,091 (37.7%)	205,783 (39.4%)
Central nervous system diseases		50,622 (5.0%)	29,657 (5.7%)
Injury		135,122 (13.4%)	54,539 (10.5%)
Jaw pain		31 (0.0%)	54 (0.0%)
General pain		4458 (0.4%)	1699 (0.3%)
Symptoms involving nervous and musculoskeletal system		43,312 (4.3%)	30,105 (5.8%)
Temporomandibular joint disorders		562 (0.1%)	336 (0.1%)
*Warfarin monitoring*			
Warfarin monitoring on current day or in prior 7 days		291,833 (29.0%)	NA

CYP = cytochrome P450; GERD = gastroesophageal reflux disease; NA = not applicable; Q = quartile. Note that the CHA2DS2-VASc score was calculated using demographic (i.e., age, sex) and healthcare claims diagnoses (e.g., hypertension, diabetes). * drugs in these subcategories with acute indications were assessed on current day or in prior 14 days. ** per Truven Health Analytics Micromedex Solutions, limited to those with “major” or “contraindicated” severity and with “good” or “excellent” documentation. † limited to clinically relevant entries in The Flockhart Table™ (Flockhart DA, Thacker D, McDonald C, Desta Z. The Flockhart Cytochrome P450 Drug-Drug Interaction Table. Division of Clinical Pharmacology, Indiana University School of Medicine (Updated 2021). https://drug-interactions.medicine.iu.edu/ (accessed on 1 March2019). ‡ cell was suppressed to maintain compliance with the Centers for Medicare and Medicaid Services cell size suppression policy (HHS-0938-2020-F-7420).

## Data Availability

Restrictions apply to the availability of these data. Data was obtained from Centers for Medicare and Medicaid Services (CMS) and are available with the permission of CMS.

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
