# Peer review of "Thromboembolic Events in Users of Warfarin Treated with Different Skeletal Muscle Relaxants"

_medicina, 2022, doi:10.3390/medicina58091171_

Round 1

Reviewer 1 Report

The authors evaluated effects of concomitant use of warfarin and individual muscle relaxants on rates of hospitalization for thromboembolism among economically disadvantaged persons. They  identified a potential >3-fold increase in the rate of hospitalized thromboembolism in concomitant users of warfarin + tizanidine vs warfarin alone.

I have the following concerns:

1. Was the study prospective?

2. Was the informed consent signed and Ethics included?

3. What was the medain CHA2DS2-VASc score?

4. Could you please define 'thromboembolic events'?

5. What was the mean time in therapeutic range?

6. What are the practical aspects of the study?

7. What was the percentage of patients with the history of thromboembolism?

Reviewer 2 Report

Leonard et al investigated the association between warfarin and skeletal muscle relaxants, and showed the potential risk of hospitalized thromboembolism in concomitant users of warfarin + tizanidine. The data is interesting and useful to understand the thrombosis/bleeding risks in anticoagulant therapy. However, there are some points to revise the contents. The details are shown below. 

1. The authors showed the confounder-adjusted incidence rate ratios and ratios of adjusted incidence rate ratios by eight kinds of muscle relaxant precipitant drug. The data is unique because the rates are largely dependent on the kind of drugs. Please describe the difference among the drugs with the drug mechanisms in the discussion section.

2. INR is used for the monitoring in warfarin therapy, and the time in therapeutic range is also important for the interpretation of warfarin monitoring data. Please add in the results section or describe the information as the limitation in the discussion section.

3. The authors showed the relationship between calendar year and the number of patients in Table 1. However, it would be better to show the relationship between calendar year and the number of patients as the separated graph.

4. The following sentence is written as the explanation of Figure 1. “Confounder-adjusted IRRs for thromboembolism ranged from 031 (95% confidence interval 0.13-0.77) for warfarin  + metaxalone to 3.44 (1.53-7.78) for warfarin + tizanidine.” Although 3.44 was written in the sentence, the figure looks like it says 3.45 in figure 1. Please confirm.

Round 2

Reviewer 1 Report

Thank you. All my concerns have been adequately addressed. I hav no further comments.